# Assessment of the Remineralizing Potential of Biomimetic Materials on Early Artificial Caries Lesions after 28 Days: An In Vitro Study

**DOI:** 10.3390/bioengineering10040462

**Published:** 2023-04-11

**Authors:** Vincenzo Tosco, Flavia Vitiello, Riccardo Monterubbianesi, Maria Laura Gatto, Giulia Orilisi, Paolo Mengucci, Angelo Putignano, Giovanna Orsini

**Affiliations:** 1Department of Clinical Sciences and Stomatology (DISCO), Università Politecnica delle Marche, 60126 Ancona, Italy; v.tosco@pm.univpm.it (V.T.); f.vitiello@pm.univpm.it (F.V.); r.monterubbianesi@univpm.it (R.M.); g.orilisi@pm.univpm.it (G.O.); a.putignano@univpm.it (A.P.); 2Department of Industrial Engineering and Mathematical Sciences (DIISM), Università Politecnica delle Marche, 60131 Ancona, Italy; m.l.gatto@univpm.it; 3Department of Materials, Environmental Sciences and Urban Planning (SIMAU) & UdR INSTM, Università Politecnica delle Marche, 60131 Ancona, Italy; p.mengucci@univpm.it; 4National Institute of Health and Science of Aging (INRCA), 60124 Ancona, Italy

**Keywords:** enamel remineralization, demineralization, early enamel lesion, SEM-EDX, CPP-ACP, nano-hydroxyapatite, F-ACP, NaF, biomaterials

## Abstract

This study aimed to evaluate the loss of mineral content in the enamel surface in early artificial lesions and to assess the remineralizing potential of different agents by means of SEM coupled with energy-dispersive X-ray analysis (EDX). The analysis was performed on the enamel of 36 molars divided into six equal groups, in which the experimental ones (3–6) were treated using remineralizing agents for a 28-day pH cycling protocol as follows: Group 1, sound enamel; Group 2, artificially demineralized enamel; Group 3, CPP-ACP treatment; Group 4, Zn-hydroxyapatite treatment; Group 5, NaF 5% treatment; and Group 6, F-ACP treatment. Surface morphologies and alterations in Ca/P ratio were evaluated using SEM-EDX and data underwent statistical analysis (*p* < 0.05). Compared with the sound enamel of Group 1, the SEM images of Group 2 clearly showed loss of integrity, minerals, and interprismatic substances. Groups 3–6 showed a structural reorganization of enamel prisms, interestingly comprising almost the entire enamel surface. Group 2 revealed highly significant differences of Ca/P ratios compared with other groups, while Groups 3–6 showed no differences with Group 1. In conclusion, all tested materials demonstrated a biomimetic ability in remineralizing lesions after 28 days of treatment.

## 1. Introduction

Remineralization is an enamel repair mechanism that can be useful for non-cavitated caries and non-carious lesions. This mechanism involves the replacement of minerals lost during the early stages of demineralization, such as calcium (Ca^2+^) and phosphate (PO_4_^3−^) ions, to restore the enamel structure and create a new surface on the rest of the existing crystals that persists after demineralization [1,2].

The first goal for dental caries prevention is to stop the ongoing demineralization process or to strengthen the remineralization process [3]. To understand the mechanism of action of remineralizing agents (RA), it is essential to comprehend the enamel’s crystalline structure and how it breaks down during the demineralization process.

Enamel hydroxyapatite (HA) crystals are highly organized units, forming compact rods that are surrounded by interrod, which extend from the dento-enamel junction to the outer surface of the tooth [4]. This extremely organized three-dimensional structure contributes to the enamel’s ability to withstand occlusal forces and counteract microbial attacks. The structure of HA crystals is under the influence of a natural cycle between demineralization and remineralization and the disruption of this balance leads to the deterioration of tooth structure [5].

The application of RA has been considered as a feasible way to prevent early enamel lesions by means of the remineralization potential exerted by the release of various ions [6]. Indeed, several functional materials have been reported to guide the deposition of minerals on the damaged enamel, thus being capable of restoring the structure and hardness of dental enamel [7].

A number of studies suggest that fluoride (F) remains the best established agent to promote remineralization [8,9]. This is attributed both to the fluoride-enhanced precipitation into the tooth enamel of Ca^2+/^PO_4_^3−^ and to the formation of fluorohydroxyapatite (F-HA) [10]. Indeed, for every two F^-^, ten Ca^2+^ and six PO_4_^3−^ are required to form one unit cell of F-HA (Ca_10_(PO_4_)_6_F_2_). Hence, with topical application of F, the availability of Ca^2+^ and PO_4_^3−^ in the oral environment can be a critical factor for the enamel remineralization process, being further intensified by xerostomic conditions [11]. F-HA is less soluble in acid solution than HA, which in turn is less soluble than carbonated apatite [12]. Therefore, original relatively acid-soluble HA is converted into relatively acid-stable F-HA. In addition, a covering CaF_2_ layer is formed on the tooth surface, which can serve as a “protective layer” during acid attacks [13].

Other non-fluoridated RA are also capable of promoting the remineralization of enamel as well as Casein Phosphopeptide–Amorphous Calcium Phosphate (CPP-ACP) and synthetic nano-hydroxyapatite (n-HA) [14]. In particular, CPPs are phosphorylated casein-derived peptides formed as the result of tryptic digestion of casein [15]; ACP is a highly reactive, non-crystalline material that rapidly converts to HA. Together, CPP-ACP nanocomplexes decrease demineralization and promote remineralization by localizing ACP in dental plaque, maintaining a state of supersaturation of Ca^2+^ and PO_4_^3−^ on the enamel surface [16,17]. In addition, the ACP can be functionalized with F, carbonate and citrate, forming the F-ACP complex, in order to efficiently remineralize damaged calcified tissues in their native structure [18,19].

In addition, n-HA has been utilized as an alternative agent for enamel remineralization for its high biocompatibility and bioactivity. Numerous studies have reported that n-HA and apatite enamel crystals possess similar properties, morphology, structure, and crystallinity [14,20,21]. Therefore, demineralized enamel could be biomimetically repaired, substituting the damaged enamel crystals with n-HA [22,23,24]. The nanosized particles can directly fill up any microporosity on the demineralized enamel surface and act as a scaffold for the precipitation of Ca^2+^ and PO_4_^3−^ from saliva in order to form a new apatite layer [25]. As a result, there is a decrease in defects and cavities of the enamel surface and an increase in hardness of its surface.

One of the main techniques for measuring the tooth’s mineral content is represented by scanning electron microscopy (SEM) coupled with energy-dispersive X-ray analysis (EDX) [26]. SEM allows obtaining qualitative information on the micromorphology of the samples because of its ability to create high-resolution images of hard surfaces; moreover, EDX represents a micro analytical technique employed to quantitatively estimate the amounts of mineral in a tooth sample [27,28].

Our previous study evaluated short-term RA effects, over a period of 7 days, demonstrating that a complete remineralization of the enamel surface was not achieved [29]. Therefore, as a continuation of this work, the aim of the present study was to evaluate for the first time the remineralizing performance of four different RA, containing CPP-ACP, n-HA, NaF, and F-ACP, respectively, on early artificial enamel lesions after a 28-day pH-cycling protocol using SEM-EDX analysis. The proposed null hypotheses were that (1) there would be no differences between the sound enamel and the enamel treated with RA in morphological features and Ca/P ratio; and (2) there would be no differences among the RA after 28 days of treatment in morphological features and Ca/P ratio.

## 2. Materials and Methods

### 2.1. Sample Preparation

The study was conducted at the Department of Clinical Sciences and Stomatology of Università Politecnica delle Marche (Ancona, Italy). Thirty-six human third molars extracted for orthodontic and periodontal reasons were collected [29]. All patients were informed in advance that their extracted teeth would be used for scientific purposes, and written consent was obtained before the experiment. For the selection, the following inclusion criteria were applied: integrity of the buccal and lingual surfaces, absence of enamel wear, traumatic lesions, and absence of volume, shape, and structural anomalies. In the selected teeth, remaining soft tissues, debris, and stains were removed with hand-scaling instruments and were stored in 0.5% *w*/*w* chloramine solution (NH_2_Cl) at room temperature. One single operator performed all procedures to avoid operator bias.

The teeth were randomly divided into six groups as follows (*n* = 6):

Group 1. Sound enamel without any treatment stored in artificial saliva (Biotene Oralbalance Gel, GSK C.Health Srl, Verona, Italy);

Group 2. Demineralized enamel as explained below in Section 2.2 and then stored in artificial saliva (Biotene Oralbalance Gel, GSK C. Health Srl, Verona, Italy);

Group 3. Artificial enamel lesion treated with a mousse containing CPP-ACP (GC Tooth Mousse, Recaldent GC, Milano, Italy);

Group 4. Artificial enamel lesion treated with a gel containing zinc-hydroxyapatite (Biorepair Desensitizing Enamel-Repair Shock Treatment, Coswell oral care professional Spa, Bologna, Italy);

Group 5. Artificial enamel lesion treated with a varnish containing sodium fluoride (NaF) 5% (Duraphat, Colgate-Palmolive, New York, NY, USA);

Group 6. Artificial enamel lesion treated with a mousse containing F-ACP (Biosmalto caries, abrasion & erosion-impact action mousse professional, Curasept Spa, Varese, Italy).

All the agents were applied once a day on the enamel surface for 120 s in a thin layer using a microbrush according to the manufacturer’s instructions.

### 2.2. Artificial Incipient Caries Like-Lesion Formation

The model for inducing artificial initial lesions on the enamel surface was used by immersing the buccal surfaces of the samples in 0.1 M lactic acid adjusted to pH 4.4 for 72 h [30]. After lesion formation, samples were thoroughly washed with deionized water, air dried, and stored in artificial saliva (Biotene Oralbalance Gel, GSK C.Health Srl, Verona, Italy).

### 2.3. 28-Day pH Cycling Protocol

The pH cycling procedure consisted of cycling in the demineralizing solution at a pH of 4.4 (0.1 M lactic acid) for 6 h (30 mL for each sample) [30,31,32,33], treating with assigned RA for 120 s, and then keeping in artificial saliva for 18 h. Between the demineralization and remineralization cycles, the teeth were washed with deionized water to eliminate the possible leftovers. The demineralizing solution was replaced every 2 days. The tested RA pH levels were as follows: Group 3 (pH 6.7), Group 4 (pH 8.0), Group 5 (pH 5.7), and Group 6 (pH 7.1). This cycle was repeated once a day for 28 days [34]. After 28 days of treatment, the teeth were carefully cleaned and dehydrated for SEM-EDX analyses.

### 2.4. SEM-EDX Analysis

Specimens were air dried, mounted on aluminum stubs, and then observed by a TESCAN VEGA 3 LMU SEM (Centre for Electron Microscopy-(CISMIN) Department of SIMAU, Università Politecnica delle Marche, Ancona, Italy). SEM images were acquired to investigate the morphology of enamel and to search for surface damage at different magnifications: 500× and 1000×.

The chemical surface characterization was performed by means of EDX using EDAX Element Microanalysis (AMETEK Gmbh, EDAX Business Unit, Weiterstadt, Germany). EDX analysis was carried out on 3 sample areas with the following operating parameters: working distance of 15 mm, acceleration voltage of 25 kV, and 500× magnification. The degree of remineralization was assessed by measuring the amount of phosphorus (P) and calcium (Ca) and calculating their ratio (Ca/P) in the treated specimens. Results were reported as mean value and standard deviation.

### 2.5. Statistical Analysis

The EDX results were analyzed using descriptive statistics, and statistical inferences between experimental groups were determined by one-way ANOVA (analysis of variance) followed by Tukey’s test, using the statistical software Prism8 (GraphPad Software, CA, USA). The group size was set to *n* = 6 for all experimental groups and significance was *p* < 0.05. The power of experiment was calculated by the G-Power software package (α = 0.05) based on preliminary evaluations of the Ca/P ratio difference between the sound enamel and the enamel treated with the four tested RA to assure that the sample size was large enough for the purpose of the test.

## 3. Results

Figure 1 and Figure 2 show scanning electron micrographs of each group (1–6), which display different enamel surface morphologies.

Scanning electron micrographs of Group 1, sound enamel without any treatment, showed the typical aspect of the intact enamel crystalline organization (Figure 1). Differently, Group 2, the demineralized untreated enamel, presented extensive surface alterations with a large dissolution of interprismatic area due to the demineralization process (Figure 2).

Unlike the previously described groups, after 28 days of treatment, enamel crystals recovery occurred in all experimental groups (Groups 3–6), even if in different ways (Figure 3, Figure 4, Figure 5 and Figure 6). The scanning electron micrographs of Group 3 displayed an almost complete surface morphology reorganization with the presence of material deposits on the surface and a slight loss of surface integrity (Figure 3). Group 4 showed enamel prisms with partially intact crystals with small areas of crystalline dissolution (Figure 4).

The SEM images of Group 5 and Group 6 highlighted the normal appearance of the remineralized enamel surface with reconstitution of the structure (Figure 5 and Figure 6).

The results of the statistical analysis of Ca, P, and Ca/P ratio assessed using SEM-EDX are shown in Table 1 and Figure 7. One-way ANOVA analysis showed that the Ca/P ratio of Group 2 was statistically different from all other groups, while there were no differences between Group 1 and all tested groups (Groups 3–6). Furthermore, there were no statistically significant differences among Groups 3–6.

## 4. Discussion

In recent years, the focus of preventive dentistry has been directed toward the non-invasive management of non-cavitated caries lesions, using RA to prevent disease progression and promote remineralization of mineral-deficient tooth structures [35]. Despite saliva representing an important source of ions for remineralization, it has a relatively low potential to remineralize early enamel lesions [36].

In recent decades, there have been efforts to develop novel biomaterials that can provide the best clinical results in this field [23,24].

Numerous techniques have been employed for the assessment of enamel remineralization [37,38,39]. Indeed, in this study, four commercially available RA were evaluated using SEM-EDX in terms of qualitative and quantitative points of view [40]. In the scientific literature, EDX is considered the gold standard for the evaluation of mineral loss or gain, measuring Ca and P elemental content in atomic percentage of sound, demineralized, and remineralized enamel surfaces. The relationship between these elements is an essential indicator of the remineralization process; thus, the Ca/P ratios for each group were calculated [41].

SEM results of Group 1, which was not treated, depict healthy enamel, homogeneously exhibiting HA crystals integrity with organized rods (Figure 1). On the contrary, Group 2 confirmed that the demineralization process led to enamel alterations with many exposed enamel prisms, depressions, and irregularities, the destruction of interprismatic spaces, and the loss of interprismatic substances, typical of enamel’s demineralization morphology (Figure 2). However, these morphological changes of the tooth surface, at the early stage, are reversible and can be repaired by the application of RA. The rationale supporting the use of these RA is their efficacy in the remineralization of early carious lesions, advocating a mini-invasive and preventive approach rather than the traditional invasive clinical ones. Indeed, the EDX results of the treated enamel of Groups 3–6 showed an increase in remineralization compared with Group 2 (*p* < 0.05). However, since Group 3 highlighted a quite high variability in results, these data should be interpreted with caution.

In general, SEM images demonstrated slight qualitative differences in enamel reorganization, peculiar to each type of tested RA. Indeed, Group 3 exhibited different surface structures compared to other groups, with some material residues and slight dissolution zones, as shown in Figure 3. This morphological non-homogeneity may imply the distribution of Ca and P elements, responsible for the higher standard deviation.

Moreover, our data from multiple comparisons revealed that there were no differences in the remineralizing effect between fluoridated RA (NaF and F-ACP) and non-fluoridated RA (CPP-ACP; n-HA). The effectiveness of F on enamel remineralization was confirmed in several studies [32,42]. This study agrees with the previous ones, as the NaF and F-ACP complex showed a good remineralizing efficacy (Figure 4 and Figure 6). F varnishes contain 5% NaF (approximately 20 times the F concentration of traditional toothpastes), which aids in the formation of long-lasting intraoral fluoride reservoirs [43]. The prolonged release of remineralizing ions over time from the RA is required to optimize the probability of caries prevention, particularly in individuals at a high risk of caries [25].

In addition, Shetty et al. claimed that NaF showed better performance than CPP-ACP in enamel remineralization, although the comparison between these two RA was statistically insignificant [34].

Furthermore, according to our results, an in vitro study showed that the n-HA and F remineralization was similar and inhibited caries development, thus suggesting that n-HA can be an effective alternative to F [44].

Based on our results, the proposed first null hypothesis, i.e., that there would be no differences between the sound enamel and the enamel treated with RA in morphological features and Ca/P ratio, was partially accepted because the morphological features appeared different, while in the quantitative analysis of Ca/P ratio no statistical differences were found. Additionally, the second null hypothesis, i.e., that there would be no differences among the RA after 28 days of treatment in morphological features and Ca/P ratio, was partially accepted because there were different morphological aspects of remineralized enamel among tested groups, while in the quantitative analysis of Ca/P ratio no statistical differences were found.

In this study, pH cycling was used to mimic the dynamics of mineral loss and gain occurring in the oral cavity, which are involved in the process of caries formation [28]. The pH cycling protocol adopted for this study was based on the model described by Featherstone et al. [45]. Various studies have performed the pH cycling process at different lengths of time ranging from 7 to 14 days [46,47,48]. In this study, a 28-day pH cycling protocol was chosen because it is believed that this was a long enough period to evaluate the potential efficacy of the RA. Indeed, Balakrishnan et al. evaluated the remineralization potential of various toothpastes over a period of 30 days and concluded that the extent of remineralization achieved was dose-dependent and increased with increasing the time of exposure and duration of the study [49]. In fact, our previous study, which evaluated the efficacy of these RA, showed that, after 7 days of treatment, a complete remineralization was not obtained; instead, only an initial reorganization of the enamel structure was observed [29].

Regarding the potential side effects, no significant differences were found between the CPP-ACP, n-HA, and fluoridated systems (NaF, F-ACP) with respect to the incidence of adverse events, including increment of dental calculus formation, allergies, and other serious side effects [50]. Conversely, the potential applications of these RA have been observed, as demonstrated by the scientific literature [6,13], to prevent and avoid damage to dental tissues. Indeed, CPP-ACP can provide effective remineralization against erosion caused by orange juice [51], increasing the hardness and density of the erosive surface of the tooth [52] and reducing the depth of lesions [53]. Furthermore, the CPP-ACP-containing mousse tested in our study may be more effective in erosion lesions due to its very low pH, in agreement with other reports [54,55]. Since F-ACP and NaF varnish are used for the prevention of enamel loss due to tooth erosion in primary teeth, parents can turn to CPP-ACP mousse or NaF-containing varnish for their children, instead of opting for less-effective fluoride-containing toothpastes. Equally valid are the applications of n-HA to remineralize and improve the physical properties of the enamel and strengthen its weakened substructure in a non-invasive way, and recover, for example, the color of white spot lesions. In fact, by releasing Ca and P ions to promote remineralization and repair demineralized areas, some studies have associated n-HA with low-concentration whitening agents, obtaining promising results for the treatment of these lesions [1,28]

Finally, the antibacterial efficacy of these RA found interesting application also in other fields of dentistry, such as in implantology, in particular against mucositis and peri-implantitis [51,56].

In the field of dental biomaterials, in vitro studies are useful for researchers to develop new materials and evaluate certain clinically relevant properties that may be difficult to evaluate otherwise. This work is the first study that qualitatively and quantitatively evaluated and compared four different RA, including CPP-ACP, n-HA, NaF, and F-ACP, with sound and demineralized enamel, for a long experimental period, i.e., 28 days. Therefore, this work can contribute to provide greater knowledge on these materials that present an increasing clinical applicability. Nevertheless, it is imperative to note that remineralization in vitro may be quite variable when compared with changes occurring in the oral cavity. Indeed, one of the study’s limitations is that an in vitro protocol and the pH cycling model are unable to completely simulate the complex intraoral conditions leading to caries development, where the pH fluctuates frequently. The oral pH levels depend upon the individual’s eating habits, oral hygiene practices, fluoride usage, and the composition and the quality of saliva and plaque [43]. For this reason, further in vitro studies and especially clinical trials are needed to validate the promising outcome of this research and clinically evaluate the remineralization efficacy of RA [57,58,59].

## 5. Conclusions

Despite the limitations of the study, according to the results obtained, it can be concluded that all tested materials had biomimetic remineralization ability on enamel subsurface lesions after 28 days of treatment. CPP-ACP, n-HA, NaF, and F-ACP highlighted good performance in terms of remineralization efficacy on early caries lesions in comparison with Group 2 (untreated demineralized enamel), even though we noted a variability in the behavior of Group 3 (CPP-ACP). Finally, all tested RA were efficient in terms of achieving good recovery of the enamel structure.

## Figures and Tables

**Figure 1 bioengineering-10-00462-f001:**
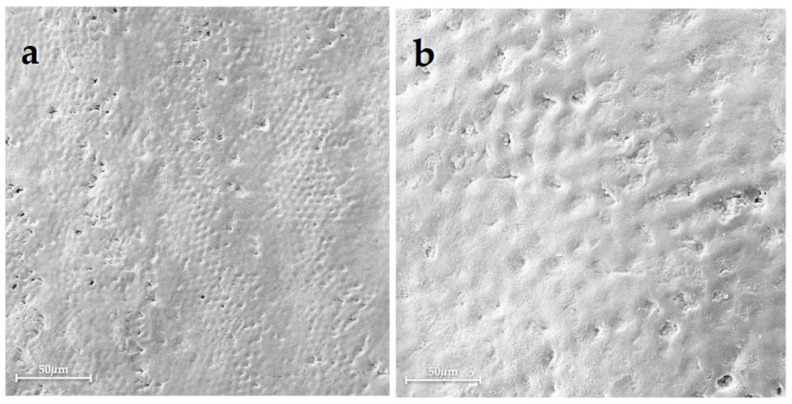
Scanning electron micrographs of the Group 1 displayed (**a**) at 500× magnification, showing the typical morphology of sound enamel with emphasized perikymatas; and (**b**) at 1000× magnification, showing slight erosion of the enamel surface.

**Figure 2 bioengineering-10-00462-f002:**
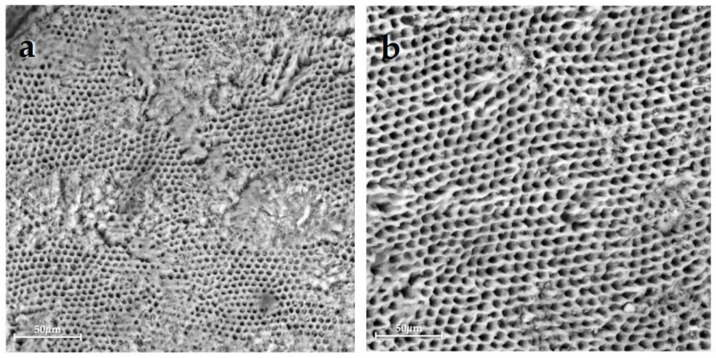
Scanning electron micrographs of Group 2 highlighted (**a**) at 500× magnification, showing the enamel surface with many exposed enamel prisms, the destruction of interprismatic spaces, and the loss of the interprismatic substance; and (**b**) at 1000× magnification, showing a honeycomb structure and a typical demineralized morphology of the enamel with the loss of the core of prisms.

**Figure 3 bioengineering-10-00462-f003:**
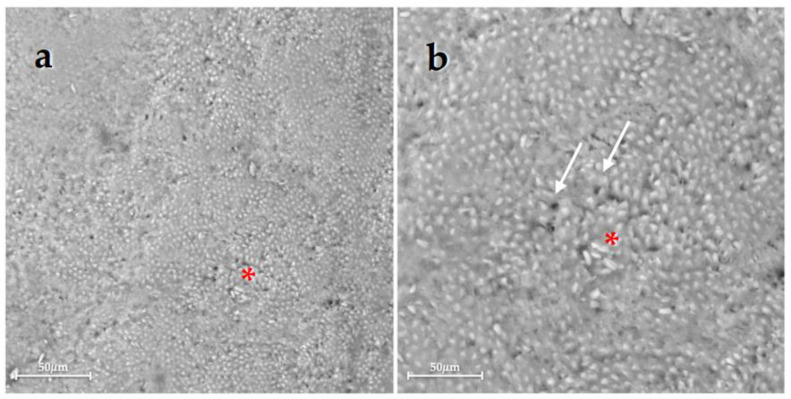
Scanning electron micrographs of Group 3 (**a**) at 500× magnification, showing an almost complete surface morphology reorganization with some material residues (red asterisks); and (**b**) at 1000× magnification, showing the presence of slightly dissoluted areas (white arrows).

**Figure 4 bioengineering-10-00462-f004:**
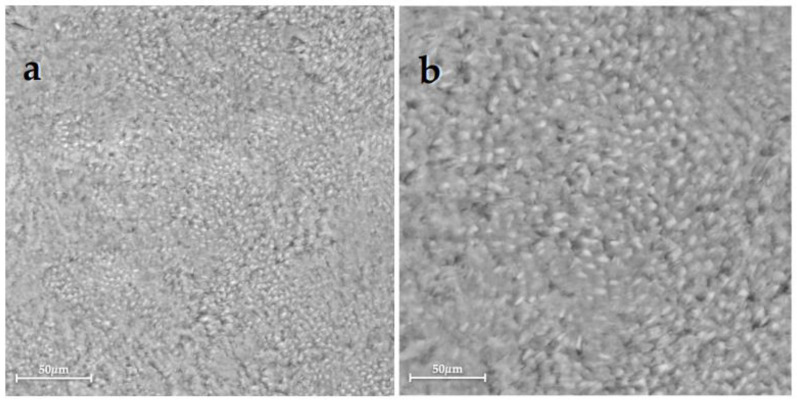
Scanning electron micrographs of Group 4 (**a**) at 500× magnification, showing the presence of a homogeneous enamel surface with an apparent structural reconstitution; and (**b**) at 1000× magnification, showing enamel prisms with partially intact crystals.

**Figure 5 bioengineering-10-00462-f005:**
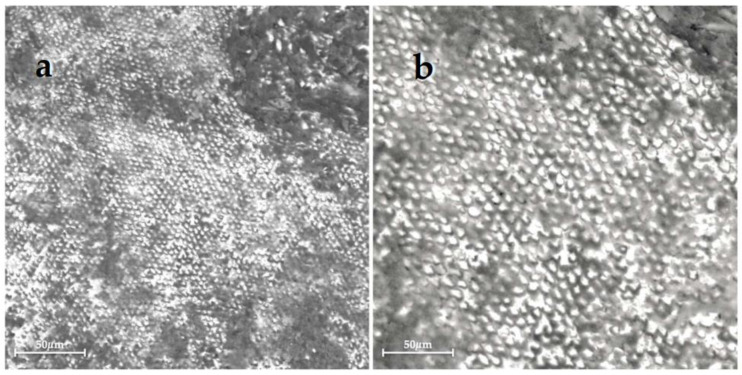
Scanning electron micrographs of Group 5 (**a**) at 500× magnification, showing the presence of a slight irregular enamel surface; and (**b**) at 1000× magnification, showing partial crystal recovery with remineralization within the prismatic structure.

**Figure 6 bioengineering-10-00462-f006:**
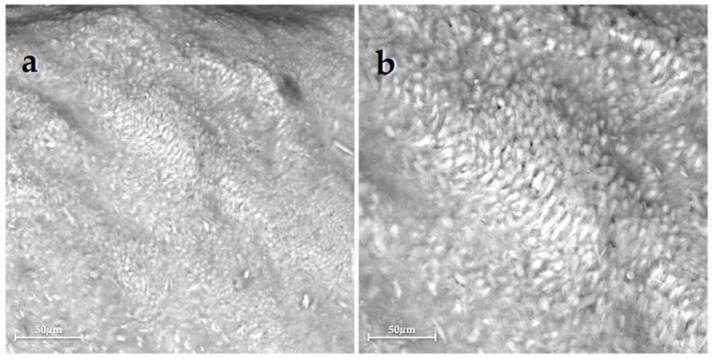
Scanning electron micrographs of Group 6 (**a**) at 500× magnification, showing a residual material covering the enamel surface; and (**b**) at 1000× magnification, showing an apparently intact interprismatic enamel structure leading to surface re-establishment.

**Figure 7 bioengineering-10-00462-f007:**
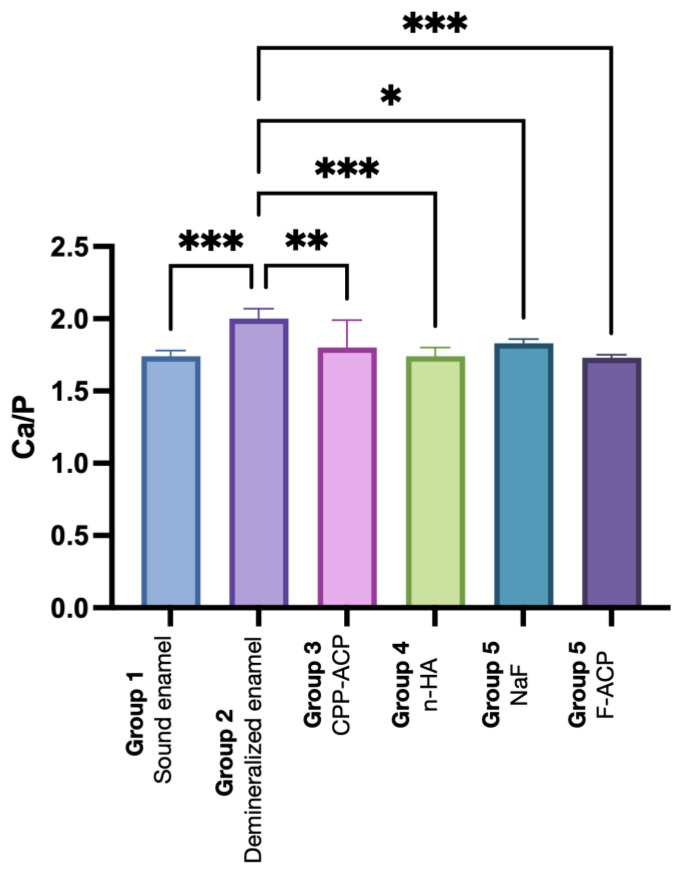
Ca/P ratio results determined using the EDX of all groups in atomic % expressed as mean and standard deviation. One-way ANOVA with Tukey’s multiple comparison test; *p* < 0.05 is significant. The following levels of statistical significance were considered: *p* < 0.05 *, *p* < 0.01 **; *p* < 0.001 ***. Comparisons between groups that are not statistically significant are not reported in the graph: Group 1 (Sound enamel), Group 2 (Demineralized enamel), Group 3 (CPP-ACP), Group 4 (n-HA), Group 5 (NaF), and Group 6 (F-ACP), respectively.

**Table 1 bioengineering-10-00462-t001:** Descriptive statistic values of the data obtained from EDX measurements of all groups in atomic % expressed as mean ± standard deviation of the Ca, P, and Ca/P ratio: Group 1 (Sound enamel), Group 2 (Demineralized enamel), Group 3 (CPP-ACP); Group 4 (n-HA), Group 5 (NaF), and Group 6 (F-ACP), respectively.

	Group 1	Group 2	Group 3	Group 4	Group 5	Group 6
Ca	63.5 ± 0.5 ^a^	66.7 ± 0.7 ^b^	64 ± 2 ^a^	63.5 ± 0.8 ^a^	64.7 ± 0.4 ^a^	63.5 ± 0.3 ^a^
P	36.5 ± 0.5 ^a^	33.4 ± 0.7 ^a^	36 ± 2 ^a^	36.5 ± 0.8 ^a^	35.3 ± 0.4 ^a^	36.6 ± 0.3 ^a^
Ca/P	1.74 ± 0.04 ^a^	2.00 ± 0.07 ^b^	1.80 ± 0.02 ^a^	1.74 ± 0.06 ^a^	1.83 ± 0.03 ^a^	1.73 ± 0.02 ^a^

^a,b^ Different superscript letters indicate statistically significant differences.

## Data Availability

The data presented in this study are available on request from the corresponding author.

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
