# Peer review of "Assessment of the Remineralizing Potential of Biomimetic Materials on Early Artificial Caries Lesions after 28 Days: An In Vitro Study"

_bioengineering, 2023, doi:10.3390/bioengineering10040462_

Round 1
Reviewer 1 Report
Dear Authors,
I have revised your work entitled “SEM-EDX assessment of the remineralizing potential of biomimetic materials on early artificial caries lesions after 28 days”. I congratulate with you for this interesting and well conducted study. Here are some suggestions to improve the manuscript. Please provide a point-by-point response, highlighting the corrections with a different color mark per each reviewer.
Title: you should add the study design in the title, that is “in vivo study”.
Abstract: The abstract counts more than 200 words. If you are able, you should try to reduce it. You can use abbreviations to reduce the length and cut some sentences (e.g. “Data were statistically analyzed for multiple group comparison by one-way ANOVA/Tukey’s test (p < 0.05)” “Data underwent statistical analysis (p < 0.05)”)
Introduction: The null hypotheses of the study should be added at the end of the Introduction and rejected/accepted in the Discussion.
Materials and methods: Sample size calculation should be added with data from previous studies. The primary outcome should be chosen (I suppose Ca/P ratio), then the calculation should be performed with expected mean, expected mean difference, standard deviation, alpha and beta errors.
Results
- You should add tables with descriptive statistics (mean, standard deviation, minimum, median, maximum) of both Ca and P alone and with Ca/P ratio. You should also add statistically significant differences between the means.
- In figure 7, you can add the acronyms of the materials after the Group n captions (eg. Group 1 – sound enamel; group 2 – demineralized enamel; group 3 – CPP-ACP; and so on).
Discussion
- A paragraph should be added on the possible implications (and advantages) of using CPP-APC, n-HA, instead of fluoride releasing products.
- Line 275 (and 282): I suggest writing “clinically” in place of “in vivo”, as in vivo refers to animal studies. Moreover, you could add a short paragraph on recent randomized clinical trials performed to test the materials used in the present study (DOI: 10.17796/1053-4625-46.1.12; DOI: 10.1111/odi.14388; DOI: 10.1007/s00784-020-03275-8).
Conclusions
- You should list all the products tested in the article.
Editorial issues
- English editing by a native speaker is recommended, spelling and editing errors should be corrected.
Author Response
Dear Authors,
I have revised your work entitled “SEM-EDX assessment of the remineralizing potential of biomimetic materials on early artificial caries lesions after 28 days”. I congratulate with you for this interesting and well conducted study. Here are some suggestions to improve the manuscript. Please provide a point-by-point response, highlighting the corrections with a different color mark per each reviewer.
Thank you for your comments, according to your suggestions, we have changed the Ms. You will find all the corrections with a different color mark for each reviewer. The changes you have requested are highlighted in green.
Title: you should add the study design in the title, that is “in vivo study”.
As we reported in the Ms, in the Materials and Methods section, we tested the remineralizing agents on 36 human third molars extracted for periodontal and/or orthodontic reasons. The experimental study was performed on initial artificial carious lesions and hence, it can be considered an in vitro study. Therefore, following your request, we added in the title “An in vitro study”.
Abstract: The abstract counts more than 200 words. If you are able, you should try to reduce it. You can use abbreviations to reduce the length and cut some sentences (e.g. “Data were statistically analyzed for multiple group comparison by one-way ANOVA/Tukey’s test (p < 0.05)” à “Data underwent statistical analysis (p < 0.05)”)
The Abstract section has been shortened (200 words) following your suggestion.
Introduction: The null hypotheses of the study should be added at the end of the Introduction and rejected/accepted in the Discussion.
Following your suggestion, the null hypotheses were added at the end of the Introduction (at lines 97-101) and then discussed in the Discussion section (at lines 273-280). The proposed null hypotheses were that (1) there were no differences between the sound enamel and the enamel treated with remineralized agents in morphological features and Ca/P ratio; and (2) there were no differences among the remineralizing agents after 28 days of treatment in morphological features and Ca/P ratio.
Materials and methods: Sample size calculation should be added with data from previous studies. The primary outcome should be chosen (I suppose Ca/P ratio), then the calculation should be performed with expected mean, expected mean difference, standard deviation, alpha and beta errors.
Thank the Reviewer for this valuable comment. We based our sample size on the first null-hypothesis that there were no differences between the sound enamel and the enamel treated with RA in morphological features and Ca/P ratio. The sample size calculation was performed using G-Power software by considering the preliminary results of the difference between the sound enamel and the enamel treated with remineralized agents in morphological features and Ca/P ratio. In particular, the effect size =1.24(calculated using the mean and standard deviation of preliminary data G1=1.68, G3=1.78; G4 1.69; G5=1.8; G6=1.72 and SD within group = 0.08) a=0.05; power (1–β) =0.80. The sample size calculation was added in the Materials and Methods section, see lines 163-167.
Results
You should add tables with descriptive statistics (mean, standard deviation, minimum, median, maximum) of both Ca and P alone and with Ca/P ratio. You should also add statistically significant differences between the means.
According to your suggestion, we added the Table 1, containing the descriptive statistical values of the data obtained from EDX measurements of all groups in atomic percentage expressed as mean ± SD of the Ca, P and Ca/P ratio, with statistically significant differences.
In figure 7, you can add the acronyms of the materials after the Group n captions (eg. Group 1 – sound enamel; group 2 – demineralized enamel; group 3 – CPP-ACP; and so on).
Following your advice, Figure 7 was changed with the description of the materials after the Group.
Discussion
A paragraph should be added on the possible implications (and advantages) of using CPP-APC, n-HA, instead of fluoride releasing products.
We have improved the Discussion section, adding a new paragraph. See lines 293-310.
Line 275 (and 282): I suggest writing “clinically” in place of “in vivo”, as in vivo refers to animal studies. Moreover, you could add a short paragraph on recent randomized clinical trials performed to test the materials used in the present study (DOI: 10.17796/1053-4625-46.1.12; DOI: 10.1111/odi.14388; DOI: 10.1007/s00784-020-03275-8).
We have changed the word “in vivo” with “clinically” and we added a new sentence regarding the clinical trials performed to test some materials used in the present study. See lines 326-328.
Conclusions
You should list all the products tested in the article.
Thanks for your suggestion. We have improved this section with conclusions regarding each material tested. See lines 332-336.
Editorial issues
English editing by a native speaker is recommended, spelling and editing errors should be corrected.
For the English revision, a native speaker colleague was asked to edit and modify the entire Ms. Her changes are in red.
Reviewer 2 Report
Rejected due to Plagiarism.
Author Response
Rejected due to Plagiarism.
We thank the Reviewer for his/her revision.
This experimental study is a continuation of the previous 7-day study in which we tested the products with a short term of use (doi:10.3390/ma15134398). Therefore, the Materials and Methods section has similarity with the previous one, as cited in ref. 29. However, the present article and the reported results are original and never published before, aimed to evaluate and compare by means of SEM-EDX the four tested products after a 28-day treatment protocol and being innovative in the field of remineralization products.
Reviewer 3 Report
The manuscript needs Major revision:
Introduction: The novelty of your work must be mentioned.
Materials & Methods: Pleases mention how did you calculate sample size?
Materials & Methods: Pleases mention the references for the time of the agent’s application time?
Results: Results of descriptive statistics of Ca/P should be demonstrated in a Table.
Discussion: Please discuss about the remineralizing potential of the studied agents from the physical and chemical points of view and search and cite more related studies and include them in your discussion.
Conclusions: Please remove the following sentences from the conclusion section:
“It is imperative to note that remineralization in vitro may be quite variable when compared to changes occurring in the oral cavity in vivo. For this reason, clinical tests must be executed to confirm our results.”
Author Response
The manuscript needs Major revision:
We thank the Reviewer for his/her useful revision. Please, find below the point-by-point response to his/her suggestions. Ms was changed accordingly. You will find in the Ms all the corrections with a different color mark for each reviewer. The changes you have requested are highlighted in blue.
Introduction: The novelty of your work must be mentioned.
As you suggest, we added the novelty of this work in the Introduction section. See lines 94-97;316-320.
Materials & Methods: Pleases mention how did you calculate sample size?
The sample size calculation was performed using G-Power software by considering the preliminary results of the difference between the sound enamel and the enamel treated with remineralized agents in morphological features and Ca/P ratio. In particular, the effect size =1.24(calculated using the mean and standard deviation of preliminary data G1=1.68, G3=1.78; G4 1.69; G5=1.8; G6=1.72 and SD within group = 0.08) a=0.05; power (1–β) =0.80. The sample size calculation was added in the Materials and Methods section, see lines 163-167.
Materials & Methods: Pleases mention the references for the time of the agent’s application time?
For the agent’s application time we based our protocol on the indications for use provided by the manufacturer of the tested remineralizing agents who recommend use for at least 120 seconds. See lines 128-129
Results: Results of descriptive statistics of Ca/P should be demonstrated in a Table.
According to your suggestion, we have added the Table 1, containing the descriptive statistical values of the data obtained from EDS measurements of all groups in atomic % expressed as mean ± SD of the Ca, P and Ca/P ratio with significant differences.
Discussion: Please discuss about the remineralizing potential of the studied agents from the physical and chemical points of view and search and cite more related studies and include them in your discussion.
We have improved the Discussion section. See lines 293-310.
Conclusions: Please remove the following sentences from the conclusion section: “It is imperative to note that remineralization in vitro may be quite variable when compared to changes occurring in the oral cavity in vivo. For this reason, clinical tests must be executed to confirm our results.”
Thanks for your suggestion, we have removed this sentence in the Conclusion section, changed, and moved it to the Discussion section, placing it as a limitation of the study. See lines 320-321.
Reviewer 4 Report
The subject of the manuscript is interesting. I think that the title of the manuscript is acceptable. The title fully describes the subject matter of the article. I believe the presented information in the manuscript is valuable and reliable. The abstract covers the most important points.
In the investigation, the author's microstructure was studied using an SEM. What parameters did you perform observation of the samples with? Add this information to the manuscript.
The manuscript's authors should necessarily refer to the other publications in the "Discussion" section. It seems to me that many interesting works about the remineralization of the enamel surface have been published in recent years, which could be added to the "Discussion" part of the manuscript. The authors can use more papers to support the work and demonstrate the knowledge behind it. I think that the discussion of the results is a little poor. The part of the "Discussion" article is similar to the report. Please expand the discussion. It is necessary to improve this part to give a complete scientific framework for the proposed research.
The above paper may be of possible technical interest. However, submitting to the Editorial review process in English of the whole paper will require considerable language revision efforts. Authors should take the help of some reliable example professional agency to proofread their manuscript or of a native English speaker with some technical knowledge of the subject.
The paper needs a slight improvement only after which it can be published.
Author Response
The subject of the manuscript is interesting. I think that the title of the manuscript is acceptable. The title fully describes the subject matter of the article. I believe the presented information in the manuscript is valuable and reliable. The abstract covers the most important points.
We appreciate the valuable feedback on our Ms provided by the Reviewer. Please, find below the point-by-point response to his/her comments. You will find in the Ms all the corrections with a different color mark for each reviewer.
In the investigation, the author's microstructure was studied using an SEM. What parameters did you perform observation of the samples with? Add this information to the manuscript.
The SEM parameters used for the observation of the samples are described in detail in subsection “2.4 SEM/EDX analysis”. See lines 145-157.
The manuscript's authors should necessarily refer to the other publications in the "Discussion" section. It seems to me that many interesting works about the remineralization of the enamel surface have been published in recent years, which could be added to the "Discussion" part of the manuscript. The authors can use more papers to support the work and demonstrate the knowledge behind it. I think that the discussion of the results is a little poor. The part of the "Discussion" article is similar to the report. Please expand the discussion. It is necessary to improve this part to give a complete scientific framework for the proposed research.
Thank you for your suggestion, we have improved and enriched the Discussion section, adding new interesting works to support and a give a more complete scientific framework for our results. See lines 251-257;293-310.
The above paper may be of possible technical interest. However, submitting to the Editorial review process in English of the whole paper will require considerable language revision efforts. Authors should take the help of some reliable example professional agency to proofread their manuscript or of a native English speaker with some technical knowledge of the subject.
For the English revision, a native speaker colleague was asked to edit and modify the entire Ms. Her changes are in red.
The paper needs a slight improvement only after which it can be published.
We hope that our changes have adequately improved the Ms.
Reviewer 5 Report
Dear authors,
the article "SEM-EDX assessment of the remineralizing potential of biomimetic materials on early artificial caries lesions after 28 days" covers an interesting topic. The introduction is clear.
Nevertheless, the research results do not add anything to the current scientific knowledge while remineralization potential of the investigated solutions is already known.
No statistical differences are detected with the control group or among the RA (the only statistical difference is the group without remineralizing agents).
Number of specimens are low (n=6), and standard deviations of some groups are too high.
No tables with values and DS are (just a graph) is available.
The authors should increase the number of specimen per group to n=10 at least.
Author Response
Dear authors,
the article "SEM-EDX assessment of the remineralizing potential of biomimetic materials on early artificial caries lesions after 28 days" covers an interesting topic. The introduction is clear.
We thank the Reviewer for his/her useful revision. Please, find below the point-by-point response to his/her comments. Ms was changed apporting the suggested modifications. You will find in the Ms all the corrections with a different color mark for each reviewer. The changes you have requested are highlighted in orange.
Nevertheless, the research results do not add anything to the current scientific knowledge while remineralization potential of the investigated solutions is already known.
The materials tested in our study, already commercially available, are currently the most used in the enamel remineralization field. We believe that the innovation of this research is in the timing of use. It represents the first work which analyzes, after 28 days of treatment, the remineralization capability of these four different active ingredients: casein phosphopeptide and amorphous-calcium-phosphate (CPP-ACP), nano-hydroxyapatite (n-HA), sodium fluoride (NaF), amorphous-calcium-phosphate functionalized with fluoride (F-ACP). Moreover, some new insights regarding the happened events during the remineralization process, in terms of morphological and chemical changes, have been exploited in this research, thus providing new advances in the knowledge that we hope could be translated to clinics. (See lines 94-97; 316-320)
No statistical differences are detected with the control group or among the RA (the only statistical difference is the group without remineralizing agents).
We agree with your comment: our results show that the only statistically significant difference in the evaluation of the Ca/P ratio, an indicator of remineralization, is between Group 2 and Groups 1,3,4,5,6. While between the RA and the control group there isn’t significant difference. We believe that this can be realistic, since the goal of these agents is to repair and recover the enamel structure lost due to the demineralization. The results of this work demonstrated that, with a long-term use (28 days), all remineralizing agents tested (CPP-ACP, n-HA, NaF, F-ACP) can reorganize the crystallin prismatic enamel surface, but these agents have demonstrated a different way of remineralizing the surface as highlighted by the scanning electron micrographs (Figs 3-6). According to your comment, we have improved the discussion section. See lines 235-257).
Number of specimens are low (n=6), and standard deviations of some groups are too high.
We thank the reviewer for his/her comment. The sample size calculation was performed using G-Power software by considering the preliminary results of the difference between the sound enamel and the enamel treated with remineralized agents in morphological features and Ca/P ratio. In particular, the effect size =1.24 (calculated using the mean and standard deviation of preliminary data G1=1.68, G3=1.78; G4 1.69; G5=1.8; G6=1.72 and SD within group = 0.08) a=0.05; power (1–β) =0.80. The sample size calculation was added in the Materials and Methods section, see lines 163-167. Furthermore, the high SD of Group 3 indicates a greater dispersion of the data around the mean value which is not linked to statistical reasons (a low of number of specimens) but precisely to the fact that the elements Ca and P in this group are less uniformly distributed than in the other groups. Indeed, Group 3 shows different surface structures from other groups with "some material residues" and "slight dissolution zones" as shown in Figure 3. This morphological non-homogeneity may imply the distribution of Ca and P elements in the sample, responsible for the higher SD. See lines 251-257.
No tables with values and DS are (just a graph) is available.
According to your suggestion, we have added the Table 1, containing the descriptive statistical values of the data obtained from EDX measurements of all groups in atomic % expressed as mean ± SD of the Ca, P and Ca/P ratio with statistically significant differences.
The authors should increase the number of specimen per group to n=10 at least.
We thank the Reviewer for this comment. We based our sample size on the first null-hypthotesis that there was no difference in Ca/P ratio between the artificial demineralized and the sound enamel. The sample size calculation was performed using G-Power software by considering the preliminary results of the difference between the sound enamel and the enamel treated with remineralized agents in morphological features and Ca/P ratio. In particular, the effect size =1.24(calculated using the mean and standard deviation of preliminary data G1=1.68, G3=1.78; G4 1.69; G5=1.8; G6=1.72 and SD within group = 0.08) a=0.05; power (1–β) =0.80. The sample size calculation was added in the Materials and Methods section, see lines 163-167.
Round 2
Reviewer 1 Report
Dear Authors,
Thank you for providing the revised version of your manuscript. The modifications performed make the manuscript suitable for publication.
Reviewer 5 Report
All the comments have been addressed